# Imatinib Mesylate Induces Necroptotic Cell Death and Impairs Autophagic Flux in Human Cardiac Progenitor Cells

**DOI:** 10.3390/ijms231911812

**Published:** 2022-10-05

**Authors:** Robert Walmsley, Derek S. Steele, Georgina M. Ellison-Hughes, Sotiris Papaspyros, Andrew J. Smith

**Affiliations:** 1School of Biomedical Sciences, Faculty of Biological Sciences, University of Leeds, Woodhouse Lane, Leeds LS2 9JT, UK; 2Centre for Human and Applied Physiological Sciences & Centre for Stem Cell and Regenerative Medicine, Faculty of Life Sciences and Medicine, Guy’s Campus, King’s College London, London SE1 1UL, UK; 3Department of Cardiac Surgery, Yorkshire Heart Centre, Leeds General Infirmary, Leeds LS1 3EX, UK

**Keywords:** receptor tyrosine kinase inhibitors, cardiotoxicity, cardiac progenitor cells, necroptosis, apoptosis, autophagy

## Abstract

The receptor tyrosine kinase inhibitor imatinib improves patient cancer survival but is linked to cardiotoxicity. This study investigated imatinib’s effects on cell viability, apoptosis, autophagy, and necroptosis in human cardiac progenitor cells in vitro. Imatinib reduced cell viability (75.9 ± 2.7% vs. 100.0 ± 0.0%) at concentrations comparable to peak plasma levels (10 µM). Imatinib reduced cells’ TMRM fluorescence (74.6 ± 6.5% vs. 100.0 ± 0.0%), consistent with mitochondrial depolarisation. Imatinib increased lysosome and autophagosome content as indicated by LAMP2 expression (2.4 ± 0.3-fold) and acridine orange fluorescence (46.0 ± 5.4% vs. 9.0 ± 3.0), respectively. Although imatinib increased expression of autophagy-associated proteins and also impaired autophagic flux, shown by proximity ligation assay staining for LAMP2 and LC3II (autophagosome marker): 48 h of imatinib treatment reduced visible puncta to 2.7 ± 0.7/cell from 11.3 ± 2.1 puncta/cell in the control. Cell viability was partially recovered by autophagosome inhibition by wortmannin, with the viability increasing 91.8 ± 8.2% after imatinib-wortmannin co-treatment (84 ± 1.5% after imatinib). Imatinib-induced necroptosis was associated with an 8.5 ± 2.5-fold increase in mixed lineage kinase domain-like pseudokinase activation. Imatinib-induced toxicity was rescued by RIP1 inhibition: 88.6 ± 3.0% vs. 100.0 ± 0.0% in the control. Imatinib applied to human cardiac progenitor cells depolarises mitochondria and induces cell death through necroptosis, recoverable by RIP1 inhibition, with a partial role for autophagy.

## 1. Introduction

Receptor tyrosine kinases are proteins critical for many cellular processes and are activated via the transfer of phosphoryl groups, driving the downstream signalling required for cell survival and proliferation [1]. Normally, this process is tightly controlled; however, when dysregulated, it can lead to carcinogenesis [1]. Consequently, over the last two decades, these proteins have been extensively investigated, leading to the development of inhibitors to prevent their activation [2]. Tyrosine kinase inhibitors are widely used drugs, with a growing range developed for targeted oncogenic properties and with great success in contributing to event-free patient survival [3]. Imatinib mesylate (IM) is a receptor tyrosine kinase inhibitor (RTKI) used to treat patients with chronic myeloid leukaemia and gastrointestinal stromal tumours. It is a relatively selective inhibitor of receptor tyrosine kinases such as c-kit, platelet-derived growth factor, intracellular Abl, and chimeric fusion protein BCR-Abl [4,5,6]. The survival of patients presenting with chronic myeloid leukaemia has greatly improved since the introduction of IM treatment [7].

Although IM has demonstrated substantial patient benefit, it has been linked to cardiotoxicity, particularly within the aging population [8]. IM may also cause cardiotoxicity manifesting in the longer term, especially in children; such problems have been shown with other chemotherapy drugs such as doxorubicin [9]. As IM is a relatively recent drug discovery, with its first clinical trial in 1998, data on such long-term manifestations have yet to be reported [10]. In an early study, reduced left ventricular ejection fraction (below 50% of baseline) was identified as a cardiotoxic side-effect in patients undergoing IM chemotherapy [11]. However, a subsequent study of 1276 cases suggested that adverse cardiac events after IM treatment were rare: 22 patients suffered congestive heart failure, with age and pre-existing cardiac conditions being contributory factors [12]. Other studies demonstrated an association of IM with hypertrophy and heart failure in patients treated for gastrointestinal stromal tumours, along with elevated levels of natriuretic peptide precursor B [13]. Further evidence of the potential adverse effect of IM was demonstrated in cardiomyocytes in vitro, which exhibited altered calcium transients, cardiomyocyte hypertrophy, mitochondrial dysfunction, and cell death [14,15,16]. Our previous study identified damaging effects of IM on adult cardiac fibroblasts, with a significant impact on viability and proliferation, alongside changes in expression of growth factors and cytokines (TGF-β1; PDGFD; IL6; IL1β [17]), suggesting that further investigation into the mechanisms of IM-induced cardiotoxicity was required, particularly in the myocyte-supporting cell populations of the myocardial interstitium.

Human endogenous cardiac progenitor cells (hCPCs) have attracted significant interest since their discovery in 2003 [18]. One endogenous hCPC population, identified by expression of the receptor tyrosine kinase c-kit (in the absence of the haematopoietic lineage marker CD45), can play a valuable role in myocardial tissue maintenance [19,20]. C-kit is a tyrosine kinase receptor expressed on the cell surface: when activated by its ligand stem cell factor, it forms a dimer, initiating the activation of cell proliferation, survival, and differentiation pathways [21]. C-kit^+^ hCPCs are capable of self-renewal and development into different myocardial cell types, including vascular smooth muscle and endothelial cells [18,19,20,22,23]. These cells have a limited ability to form cardiomyocytes, although the extent to which they can do this, particularly in ischaemic tissue, is highly debated [18,22,24,25]. These cells can, however, contribute to cardiac tissue maintenance via both the replacement of damaged vascular cells and by aiding recovery of injured myocytes through the release of pro-survival growth factors [19,23]. Cardiac progenitor cells have also been shown to be necessary for adult myocardial regeneration in a rodent model of diffuse myocardial injury, involving cardiomyopathy induced by catecholamine excess and removal of proliferating CPCs by the antimitotic agent 5-flurouracil. This combination led to severe heart failure, with functional recovery dependent upon CPCs [19]. Therefore, hCPCs may play a key role in resistance to, or recovery from, RTKI-induced cardiotoxicity by providing protection to cardiomyocytes at risk of a comparably diffuse injury: RTKI-induced cell death.

To fully understand how RTKIs cause adverse effects on the myocardium, it is important to investigate their impacts on cell death pathways. Apoptosis is a tightly regulated process defined by the activation of pro-apoptotic factors such as executioner caspases 3 and 7, which eventually lead to DNA fragmentation and cell death [26]. Necroptosis (programmed necrosis) is an alternative cell death pathway regulated by the expression of activated proteins: receptor-interacting serine/threonine-protein kinase 1 (RIP1), receptor-interacting serine/threonine-protein kinase 3 (RIP3), and mixed lineage kinase domain-like pseudokinase (MLKL) [27,28,29]. Another relevant cellular process is autophagy, which has been linked to cell death, including the leakage of pro-cell death signals from lysosomes [30]. However, other studies have shown autophagy to be an initiator of different cell death pathways such as apoptosis, albeit not their direct cause [31,32]. Understanding the involvement of autophagy in IM-induced cell death could have significant clinical importance as autophagy has been proposed as a cause of IM resistance in cancer patients [33]. The present study addresses the cell death pathways activated by IM in hCPCs with a view to identifying possible strategies to overcome this toxicity.

## 2. Results

### 2.1. IM Reduces Cell Viability but Does Not Induce Apoptosis

To determine whether RTKIs caused cell death in the hCPC population, cells were grown in vitro and exposed to a range of concentrations of IM, and cell viability was determined by fluorescein diacetate (FDA) staining. Cell viability decreased by 25% relative to the control (*n* = 5, *p* < 0.05) after the cells were treated with IM for 24 h (Figure 1A). Following this, the mechanism of IM-induced toxicity was investigated to determine whether cell death was occurring through apoptosis or an alternative pathway. ToPro-3 staining was apparent within the nuclei of hCPCs after treatment with IM (representative images in Figure 1B,C). There was a dose-dependent increase in ToPro-3-positive cells after IM treatment, with a significant increase of 13.6 ± 1.5% after 100 µM IM (*n* = 4, *p* < 0.05) (Figure 1D). Neither control cells nor IM-treated cells showed increased caspase 3/7 activity (Figure 1B,C). To reaffirm that IM-induced cell death was not apoptosis, we performed Annexin V labelling of treated cells, with flow cytometry analysis of this confirming this. Untreated cells were 2.27% Annexin V-positive (Figure 1E), compared to 95.53% after 24 h of exposure to 10 µM staurosporine (Figure 1F), but only 2.05% after 24 h of treatment with 10 µM IM (Figure 1G) or 2.82% with 100 µM IM (Figure 1H). Real-time RT-qPCR was used to detect changes in gene expression after exposure to 10 µM IM for 24 h. There were no significant changes in expression of either apoptotic (BAX, caspase 8, PARP, calpain, and TNF-α) or necroptosis-associated genes (RIP1, RIP3, and MLKL). The largest expression of fluctuations seen in apoptosis-associated genes were in caspase 8 (1.6 ± 0.5-fold) and TNF-α (1.6 ± 0.2-fold) (*n* = 3) (Figure 1I).

### 2.2. IM Impairs Mitochondrial Membrane Potential in hCPCs

There was a reduction in the number of cells with high-intensity TMRM staining after 10 µM IM treatment, with 20% of these cells having a reduced intensity; 20 µM of IM reduced the number of cells with high intensity by 38% and 50 µM IM reduced the number of such cells by 60% (Figure 2A–D). Finally, the positive control trifluoromethoxy carbonylcyanide phenylhydrazone (FCCP) decouples the mitochondria, reducing by 96% cells within the gated region relative to untreated control cells (Figure 2E). These data are reinforced by the changes in mean fluorescent intensity (MFI) of TMRM, with a reduction in average fluorescence of 74.6 ± 6.5% after 10 µM IM treatment and of 49.2 ± 6.8% after 50 µM IM treatment, relative to the control (*n* = 6, *p* < 0.05) (Figure 2F).

### 2.3. IM Impairs Autophagic Flux in hCPCs

The cells exposed to 10 µM IM over periods of 24, 48, and 72 h showed a significant increase in the number of acidic organelles (Figure 3A–D). Quantification of acridine-orange-positive cells showed control cells to be 9.0 ± 3.0% positive, IM-24 h 55.5 ± 6.4% positive, IM-48 h 60.0 ± 5.7% positive, and IM-72 h 73.2 ± 3.1% positive (Figure 3E). To examine underlying mechanisms, Western blotting was used to analyse the expression of lysosomal Associated Membrane Protein 2 (LAMP2, a specific lysosomal protein marker). An increase in LAMP2 protein expression was seen in Western blotting, quantified through densitometry, with a 2.4 ± 0.3-fold increase in LAMP2 protein expression after exposure of cells to 10 µM IM for 24 h (Figure 3F).

Having identified that IM increased the density of acidic organelles and lysosome formation, it was important to evaluate the autophagic flux. Autophagic flux can be defined as how well autophagosomes can fuse with lysosomes, in order to remove unwanted proteins such as p62. To measure autophagic flux, Western blotting was used to analyse the lipidation of the autophagosome marker LC3I to LC3II and the removal of the P62 protein. A lysosomal inhibitor known as bafilomycin A1 (bafilomycin) was used to prevent protein degradation; therefore, if the cell had a functional autophagic flux, there would be an increase in protein levels after bafilomycin inhibition. Western blot analysis of 10 µM IM treatment for 24 h caused 2.7 ± 1.3- and 1.9 ± 0.0-fold increases in LCII and P62, respectively (*p* < 0.05), which further increased with bafilomycin to 3.5 ± 0.3- and 3.9 ± 0.9-fold increases (*p* < 0.05). However, 10 µM IM treatment for 48 h caused 2.6 ± 0.2- and 2.8 ± 0.3-fold changes of LC3II and P62, respectively, with no significant increases following bafilomycin co-treatment: 2.7 ± 0.1 and 2.6 ± 0.0 relative to the untreated control cells (1.0 ± 0.0) (Figure 4A,B).

These data were reaffirmed using the PLA assay: this enables a resolution of 50 nm, giving more detailed images to confirm the colocalisation between LC3II and p62. Cells were left untreated (Figure 5A) or treated with 10 µM IM for 24 or 48 h (Figure 5B,C), then fixed and stained for LC3II and LAMP2. Cells treated with rapamycin for 4 h provided a positive control for autophagy (Figure 5D). A clear increase in red puncta density was seen when examining control cells and rapamycin; more puncta were also seen when comparing the control to 10 µM IM treatment for 24 h. However, there was a clear reduction in red puncta when comparing control cells with those treated by 10 µM IM for 48 h. These appearances were confirmed by quantification of PLA puncta per cell: control cells showed 11.3 ± 2.1 puncta per cell, which increased to 32.1 ± 3.7 after 10 µM IM treatment for 24 h (*p* < 0.05). The puncta were significantly decreased in number after 10 µM IM treatment for 48 h: 2.7 ± 0.7 compared to levels after 24 h (Figure 5E) (*n* = 10, *p* < 0.05).

### 2.4. Autophagic Impairment Is Not the Main Contributor to IM-Induced hCPC Death

Having determined that IM impairs autophagic flux, it was important to identify whether this contributes to cell death. To examine this, an autophagosome inhibitor was used (wortmannin), which prevents the initial formation of autophagosomes and is, therefore, an upstream autophagy inhibitor. Western blotting was carried out to ensure that wortmannin was preventing autophagosome formation, by analysing LC3II expression (Figure 6A). Densitometry analysis of LC3II levels identified a decrease in LC3II expression after co-treatment with 10 µM IM and 200 nM wortmannin for 24 h: 1.5- vs. 2.2-fold when cells were treated with IM only. There was no change when cells were treated with IM for 48 h and co-treatment with wortmannin: 2.5- vs. 2.3-fold IM only (Figure 6B). Due to limits of available hCPCs, this experiment could be performed only once. Following this, cell viability was measured using the FDA assay on treated and control cells. IM treatment for 24 h reduced the cell viability to 84 ± 1.5% (*p* < 0.05 relative to untreated control), whereas in cells co-treated with IM and wortmannin for 24 h, cell viability decreased to 91 ± 3.7% (Figure 6C).

### 2.5. IM Induces Necroptotic Cell Death within hCPCs, Reversed by RIP1 Inhibition

Western blotting revealed a clear increase in total MLKL and phosphorylated MLKL after IM treatment (Figure 7A). Densitometry of phosphorylated MLKL confirmed this, with an 8.5 ± 2.5-fold increase after 10 µM IM treatment compared to the control (Figure 7B). Necrostain-1 (nec-1) was used to inhibit the upstream necroptosis initiator RIP1; following nec-1 inhibition, cell viability was measured. Compared to the control, 10 µM IM treatment for 24 h reduced the cell viability to 78.2 ± 2.5% (*p* < 0.05), whereas 10 µM IM treatment for 24 h co-treated with nec-1 reduced the cell viability to 88.7 ± 3.1% (*n* = 3, *p* > 0.05) (Figure 7C).

## 3. Discussion

This study demonstrates that IM causes significant impacts on the hCPC population. Only a few previous studies have fully investigated the cell death pathways induced by IM, with a very small number addressing IM effects on hCPCs [17,34,35,36]. Here, we found that IM reduced hCPC viability, but without inducing apoptosis. Previous reports have shown that IM induced an increase in caspase 3 and 7 expression, BAX, and cytochrome C and TUNEL-positive marking, in cardiomyocytes and cancer cell lines [11,35]. This present study did not find evidence of activated caspase 3 and 7, or increased expression of apoptosis-associated genes. However, there was an increase in ToPro-3 staining, consistent with severe plasma membrane damage and cell death. Further investigation of the IM effects on the mitochondrial membrane potential is consistent with previous findings in cardiomyocytes and chronic myeloid leukaemia cells, with IM causing a reduction in membrane potential [11,37,38]. One reason for this could be leakage of lysosomal content such as zinc during autophagy, as previously shown by Li et al. [39]. Therefore, it was important to analyse the contribution of autophagy in IM-induced cell death. A possible alternative link could be that IM may act via activation of the ADP/ATP carrier, akin to that seen by FCCP, as used for reference in our study, but this seems unlikely. This effect was dependent on both the ADP/ATP carrier and on uncoupling protein 1, and other work in primary adipocytes showed that there was no change in uncoupling protein 1 gene expression following IM treatment [40].

The number of acidic organelles increased after IM application, which was accompanied by increased expression of the lysosomal structural protein LAMP2. This is similar to findings in previous studies in the mouse neuroblastoma cell line N2a, with one study suggesting IM-induced autophagy is due to inhibition of c-Abl rather than of c-kit or PDGF [41]. There are, to our knowledge, no previous studies investigating autophagic flux in hCPCs. Here, autophagic flux was shown to be impaired, so although IM increased the amount of autophagosomes and lysosomes, their ability to fuse was likely impaired. However, when IM-induced autophagy was blocked by the upstream autophagy inhibitor wortmannin, there was no significant decrease in cell viability compared to untreated cells, suggesting no toxic effect on the CPCs. Wortmannin could, therefore, be a potential therapeutic target to aid in the protection of hCPCs, although the existing data are contradictory. One previous study showed that inhibiting upstream autophagy using 3-methyladenine (3MA) or Atg5 inhibition impaired the chemotherapeutic effect of IM in a range of cancer cell lines [42]. In apparent contradiction, other reports using K562R cells have shown 3MA improved IM-induced cancer cell death [43]. Therefore, further work is required to understand fully if wortmannin could be used to protect hCPCs, while also avoiding inhibition of IM’s chemotherapeutic action.

Due to the impairment of autophagic flux and an increase in the p62 protein, this study investigated the involvement of necroptotic cell death in IM-induced toxicity. A previous study by Goodall et al. showed the presence of the p62 protein to be vital for MLKL phosphorylation; if p62 was removed, the cell death pathway would switch to an apoptotic cell death [44]. This current study has shown the absence of key apoptotic markers after IM treatment: the expression of phosphorylated MLKL was, therefore, measured to determine whether IM was inducing cell death through necroptosis. We found an increase in MLKL activation compared to untreated cells. This cell death pathway has yet to be identified within IM-induced cardiotoxicity or within previous cancer cell research. These previous studies have shown IM-induced toxicity to occur through apoptotic cell death [11,36,45,46], although one study was unable to identify apoptosis in cardiomyocytes [14]. As the viability of hCPCs was rescued using the RIP1 inhibitor nec-1, this could provide a valuable therapeutic target to protect hCPCs from damage and thereby support cardiomyocyte survival. Further experiments would be needed to ensure nec-1 is not involved in the chemotherapeutic action of IM, although previous cancer studies show cell death through apoptosis and not necroptosis; this could, however, be due to a lack of investigation into necroptosis mechanisms.

As this study was aimed at characterising the mechanisms underlying the toxicity of IM to CPCs, following our previous study of the broad impacts of IM and two other RTKIs on the CPC phenotype [34], we also did not investigate the contribution of pharmacologically active IM metabolites to cell death mechanisms. Another step towards more precise replication of the clinical cardiotoxicity context would be to use clinical-trial-purity IM, with both steps further advancing our findings in CPCs.

In summary, this study demonstrated a range of impacts caused by the clinically used cardiotoxic drug IM on human endogenous cardiac progenitor cells. The study identified two possible targets to overcome IM-induced toxicity on hCPCs, through either the inhibition of upstream autophagy or inhibition of necroptosis (Figure 8).

## 4. Materials and Methods

### 4.1. Cell Culture

All cell culture techniques were performed under aseptic conditions. Growth medium was composed using two solutions. Solution A comprised DMEM-F12-Ham containing insulin-transferrin-selenium (1% *v*/*v*), basic-FGF (10 ng/mL), EGF (20 ng/mL), and human leukaemia inhibitory factor (10 ng/mL). Solution B comprised Neurobasal medium supplemented with Glutamax (2% *v*/*v*), B27 supplement (2% *v*/*v*), and N2 supplement (1% *v*/*v*). To prepare growth medium: 45% of solution A and 45% of solution B were combined and supplemented by: embryonic-stem-cell-qualified FBS (10% *v*/*v*); penicillin-streptomycin (1% *v*/*v*); Fungizone (0.1% *v*/*v*); gentamicin (0.1% *v*/*v*), and then sterilised through a 0.22 μm pore filter. ‘Medium’ is defined as above, unless otherwise stated.

### 4.2. Human Cardiac Progenitor Cell Isolation

Adult hCPCs were isolated from myocardial tissue obtained during the routine course of cardiac surgery. Briefly, biopsies were weighed and dissected into 0.5–1 mm^3^ pieces and subjected to collagenase type II digestion 0.3 mg/mL (Lorne Laboratories, Earley, United Kingdom) at 37 °C: first, for 5 min, followed by 7–9 repeat digestions for 3 min, with each post-digest suspension passed through a 100 µm filter. After tissue sample digestion, all collected cells were strained through a 40 µm filter to isolate the smaller cell population. The collected cell fraction was spun at 400× *g* for 10 min, prior to elimination of debris through Optiprep solution (Sigma, St. Louis, MO, USA). Cells were pelleted by centrifugation through a dual-layered Optiprep suspension: lower layer of 36% (16 mL) and upper layer of 16% (16 mL) at 800× *g* for 20 min. The cells were then negatively selected for CD45 and positively selected for CD117 (c-kit) using magnetic bead separation (Miltenyi Biotec, Bergisch Gladbach, Germany). After selection, the hCPCs were re-suspended in medium and plated onto a coated 6-well plate for conditioning. Cells were cultured in a humidified tissue culture incubator at 37 °C, 5% CO_2_, and 1% O_2_ and passaged upon reaching 70–80% confluency. All reagents were sterilised through a 0.22 µm filter.

### 4.3. Imatinib Treatment

Imatinib (Synkinase, San Diego, CA, USA) was either added at the peak-plasma-level-equivalent concentration of 10 µM (for reverse-transcription qPCR, acridine orange staining, proximity ligation assay, and Western blotting experiments), or in a dose-dependent manner of 1–100 µM according to experimental conditions (cell viability, mitochondrial membrane potential staining, and live cell staining for high-content image analysis). This study used RTKI concentrations chosen after review of relevant study data for clinical plasma concentrations at peak and trough [47,48] levels. Our work also used higher concentrations to determine any dose-related impacts (although 100 µM is well above clinical plasma levels).

### 4.4. Cell Viability

The FDA cell viability assay was performed as previously described [49]. Briefly, 5 × 10^3^ hCPCs were plated into each well of a 96-well plate and treated with IM in medium for 24 h. After treatment, medium was removed and replaced with 5 µg/mL and FDA (in 1% ethanol, 9% PBS, and 90% DMEM-F12), and incubated for 10 min at 37 °C. Fluorescent readings were taken using a Varioskan Flash plate-reader (v.4.00.53, Thermo Fisher, Waltham, MA, USA) at excitation/emission wavelengths of 485/520 nm. Cell viabilities of treatment groups were calculated as a percentage of the untreated control group. For cell viability with the autophagy inhibitor, cells were left to attach for 24 h before 10 µM of IM treatment and co-treatment with 200 nM wortmannin for 24 and 48 h time points, before applying FDA as described.

### 4.5. Live Cell Staining

Live cell staining for high-content image analysis: cells were added to a CellCarrier-96 Ultra Microplates (Perkin Elmer, Waltham, MA, USA), 1000 cells/well. Fam/Cas dye (1:150) was added and incubated for 1 h and washed with apoptosis wash buffer (Life Technologies V35118; diluted 1:10). Hoechst 1:200, ToPro-3 1:1000, and TMRM 1:100,000 (Invitrogen, Waltham, MA, USA) were added and incubated for 10 min, then replaced with medium (without indicators) and analysed by an Operetta platform (Perkin Elmer). Results were interpreted using the Columbus™ software (Perkin Elmer). Annexin staining was carried out with an Annexin V FITC labelling kit (Miltenyi Biotec), analysed by a Cytoflex S flow cytometer (Beckman Coulter, Indianapolis, IN, USA) using a 488/525 nm filter. 

### 4.6. Mitochondrial Membrane Potential Staining

To evaluate the effect of IM on mitochondrial membrane potential, flow cytometry was used to analyse the retention of a cell-permanent cationic dye called tetramethylrhodamine methyl ester perchlorate (TMRM). Cells were seeded onto 100 mm tissue culture dishes until 80% confluent; TMRM (1:300,000) was added for 10 min and the cells were then washed with PBS before being detached using Accutase (Sigma). Following this, the cells were centrifuged at 300× *g* for 5 min and then re-suspended in a PBS wash; the cells were then pelleted at 300× *g* for 5 min. Finally, cells were re-suspended in incubation buffer: PBS (Ca^2+^ and Mg^2+^-free, Invitrogen 14190-136); 2.5 g of bovine serum albumin (Sigma A9418); 1% (*v*/*v*) Penicillin/Streptomycin (Invitrogen 15140-122); 0.1% (*v*/*v*) Fungizone (Invitrogen 15290-018); 0.1% (*v*/*v*) Gentamicin (Sigma G1397), and analysed using a Cytoflex S flow cytometer using a 561/585 nm filter. Data were interpreted using Cytexpert and Excel. The mitochondrial membrane potential was calculated using the fluorescence intensity of TMRM of treated cells relative to untreated cells. 

### 4.7. Acidic Organelle Staining

To determine the role of autophagy in IM-induced cell death, acridine orange (ImmunoChemistry Technologies, Davis, CA, USA) was used to determine if the drug increased acidic properties within the cell (acidic organelles), indicating the presence of increased lysosome content. Cells were plated into a 12-well plate and allowed to reach confluency. Following this, 5 µM acridine orange was added to medium for 30 min. Cells were then washed once with PBS and imaged using confocal microscopy (LSM880, Zeiss, Oberkochen, Germany). The number of positive cells were counted using Image J (National Institutes of Health, Bethesda, MD, USA).

### 4.8. Reverse Transcription Quantitative PCR (RT-qPCR)

Cells were plated onto 100 mm dishes and treatments added at a clinically comparable concentration. Cells were pelleted and re-suspended in 1 mL of PBS with antibiotics (1% (*v*/*v*) Penicillin/Streptomycin (Invitrogen 15140-122); 0.1% (*v*/*v*) Fungizone (Invitrogen 15290-018); 0.1% (*v*/*v*) Gentamicin (Sigma G1397)) and centrifuged at 1500 rpm for 5 min. Briefly, RNA was then isolated using the QIAshredder and RNeasy mini kit (Qiagen, Hilden, Germany), according to the manufacturer’s guidelines. The RNA sample purity was tested using a Nanodrop 2000 (Thermo Fisher). The iScript cDNA synthesis kit and BioRad (Hercules, CA, USA) CFX96 Real-Time system were used to generate 100 µL of cDNA from 2000 ng of RNA. mRNA sequences for all primers were obtained from the NIBC nucleotide database and assessed using Primer-BLAST software (National Center for Biotechnology Information, Bethesda, MD, USA). All PCR primers (Table 1; Sigma) were reconstituted in molecular-grade water (Sigma) to produce 100 µM long-term stocks and 5 µM working stocks. SYBR green (BioRad), molecular-grade water (BioRad), and forward and reverse primers were added to a 96-well plate, with 2 µL of cDNA per well added to give a total volume of 20 µL per well. Samples were run for 40 thermal cycles with real-time imaging of PCR product generation (CFX96, BioRad). Data were analysed using CTX manager software (BioRad) and Excel (Microsoft, Redmond, DC, USA), with data representing ∆∆Cq values.

### 4.9. Proximity Ligation Assay

The proximity ligation assay (PLA) gave visualisation of two proteins within 40 nm of each other (indicating higher probability of direct binding). The Duolink kit was used (Sigma DUO92002, DUO92004, and DUO92008) with cells cultured on 8-well glass chamber slides (Thermo Fisher 155409) and treated with 10 µM IM for 24 or 48 h, or 400 nM rapamycin for 4 h. Cells were fixed with 2% paraformaldehyde for 10 min at room temperature, permeabilised with 0.1% Triton X-100, and blocked with goat serum (Thermo Fisher, 10000C) for 1 h at room temperature. Cells were then incubated with primary antibodies: 1:200 LAMP2 (Novus) and 1:200 MAP1LC3B Antibody (CSB- Cusabio PA887936LA01HU), overnight at 4 °C. For secondary incubation, PLA (+) and PLA (−) probes were incubated for 1 h in a humidified incubator. For each treatment, 8 µL of PLA (+) and 8 µL of PLA (−) were added in 24 µL of blocking buffer and placed onto treatment wells. Cells were washed with PLA wash buffer A and incubated with ligation reaction (40 µL: 8 µL ligation buffer; 1 µL ligase; 31 µL dH_2_O) for 30 min in a humidified chamber. Cells were washed 3 times with PLA wash buffer A and incubated with amplification reaction (40 µL: 8 µL amplification buffer; 0.5 µL DNA polymerase; 31.5 µL dH_2_O) for 100 min in a humidified incubator at 30 °C. Finally, cells were washed 3 times with PLA wash buffer A and 0.01% wash buffer B for 10 min. Results were visualised by confocal microscopy (LSM880, Zeiss). Following image acquisition, PLA puncta staining was counted manually for each condition. 

### 4.10. Western Blotting

Cells were plated on 100 mm tissue culture dishes and treated with: 10 µM IM for 24 h; 400 nM bafilomycin for 3 h; 10 µM IM for 24 h; 400 nM bafilomycin for the last 3 h of treatment; untreated aside from medium change (control). For the necroptosis-positive control, cells were incubated with 20 ng/mL of TNF-α (Peprotech, Cranbury, NJ, USA), 100 nM SMAC mimetic Birinapant (BioVision, Waltham, MA, USA), and 20 µM V Z-VAD(OMe)-FMK (Tonbo biosciences, San Diego, CA, USA) for 8 h. After treatment, cells were lysed using RIPA buffer (Sigma) and quantified using the Pierce™ Rapid Gold BCA Protein Assay Kit (Thermo Scientific). Protein was loaded (10–50 µg per well, depending on protein of interest) and transferred using a semi-dry turbo blotter (Bio-Rad). Blots were placed into 5% milk solution (or 5% bovine serum albumin, BSA, for phosphorylated proteins) containing primary antibodies and were incubated overnight at 4 °C (Table 2). The blots were then washed three times (10 min per wash) in TBS-Tween; following this, the blots were placed in 5% milk solution containing the relevant secondary antibody, either anti-rabbit (Cell Signaling) or anti-mouse (Cell Signaling, Danvers, MA, USA), at 1:1000 dilution for 1 h at room temperature.

Blots were developed using West Pico plus (Thermo Fisher) and imaged on the automated developer G-box (Syngene, Cambridge, UK). Densitometry was analysed using Image J and Excel, with treatments normalised against control groups.

### 4.11. Statistical Analysis

One-way ANOVA was used for comparison of mean data between groups for analyses of cell viability, live cell staining, mitochondrial membrane potential staining, and PLA assay, followed by the post hoc Tukey test for comparisons between groups. A two-tailed, unpaired Student’s *t*-test on ∆∆Cq values was employed to compare the target gene expression between each treatment and the respective control values.

## Figures and Tables

**Figure 1 ijms-23-11812-f001:**
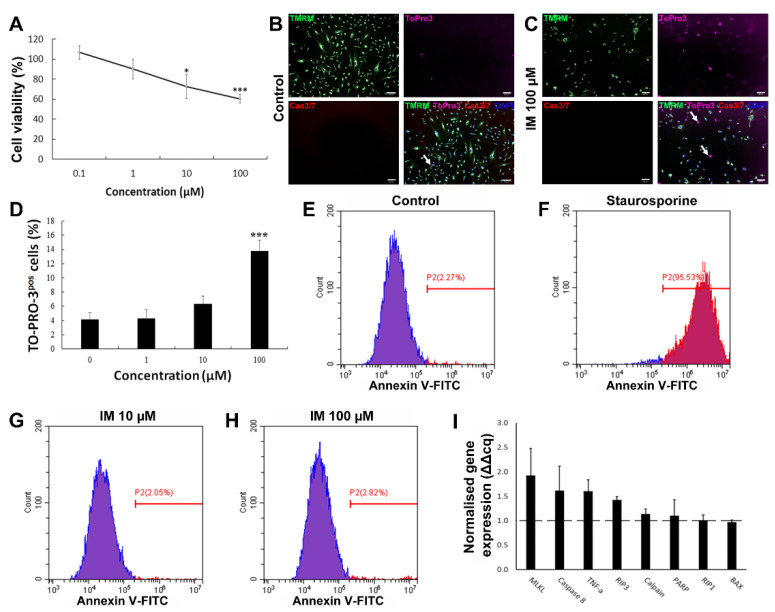
IM reduces hCPC viability but does not induce apoptosis. (**A**) Cell viability as percentage of untreated control, IM treatment range: 0.1–100 µM, *n* = 5. (**B**,**C**) Representative images of live cell staining with mitochondrial marker TMRM and nuclear marker DAPI, with ToPro-3 (arrows indicating selected dead cells) and caspase 3 and 7 substrate; IM application caused no caspase activity. (**D**) Quantification of ToPro-3 shows increased ToPro-3 staining with increased IM concentration, *n* = 4. (**E**–**H**) Flow cytometry analysis of Annexin V staining confirms IM does not induce apoptosis (Annexin V negative in blue, positive in red). (**I**) Real-time RT-qPCR shows no apoptosis-associated gene expression change, *n* = 3, control value = 1 (dashed line). Data are mean ± SEM; * *p* < 0.05 and *** *p* < 0.001 vs. control. Scale bars = 100 µm.

**Figure 2 ijms-23-11812-f002:**
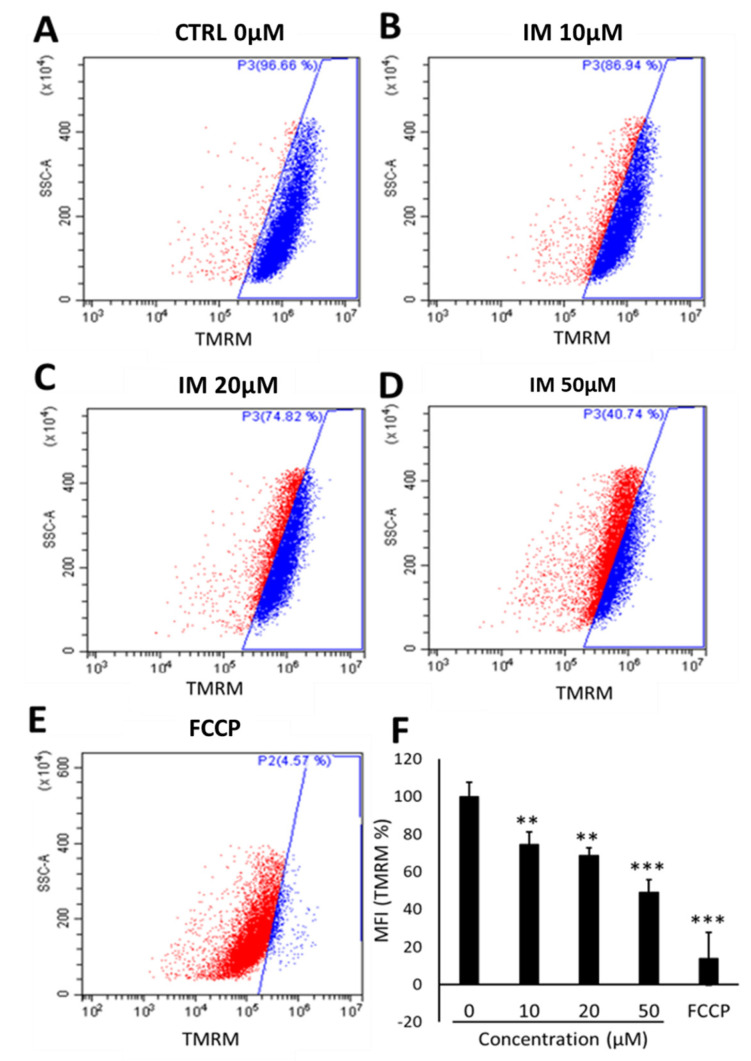
IM reduces hCPC mitochondrial membrane potential. Changes in TMRM fluorescent intensity with increased concentration of IM: (**A**) Control (blue), (**B**) 10 µM IM with slight left shift in intensity (red), (**C**) 20 µM IM, (**D**) 50 µM IM, and (**E**) positive control of 10 µM FCCP. (**F**) Quantification of mean florescence intensity (MFI) of TMRM normalised against control (100%). Data are mean ± SEM, *n* = 5; ** *p* < 0.01 and *** *p* < 0.001 vs. control.

**Figure 3 ijms-23-11812-f003:**
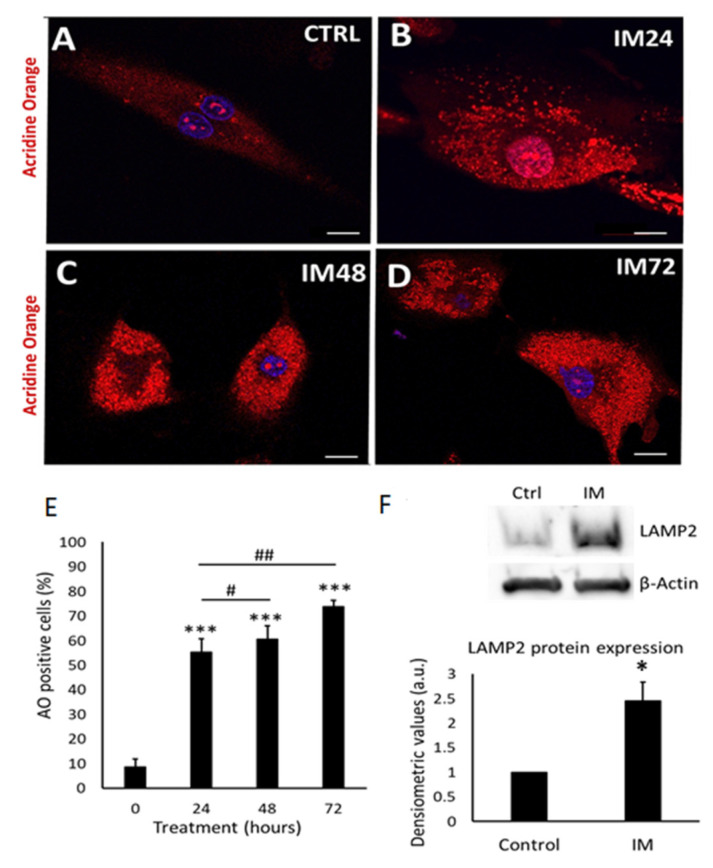
IM induces accumulation of acidic vesicle organelles, indicative of autophagy and LAMP2 expression. (**A**–**D**) Representative images of hCPCs stained with acridine orange; orange staining represents acidic organelles: control cells and IM (10 µM) for 24, 48, and 72 h treatments. (**E**) Quantification of acridine orange staining in hCPCs after IM treatment (10 µM) relative to control. Data are mean ± SEM for percentage of acridine-orange-positive cells, *n* = 7. (**F**) Western blot image for LAMP2, control cells, and IM-treatment for 24 h; beta-actin represents loading control. Densitometry for LAMP2 Western blots average signal intensity normalised against control. Data are mean fold change ± SEM, *n* = 3; * *p* < 0.05 and *** *p* < 0.001 vs. control; # *p* < 0.05 and ## *p* < 0.01 vs. IM 24 h. Scale bars = 20 µm.

**Figure 4 ijms-23-11812-f004:**
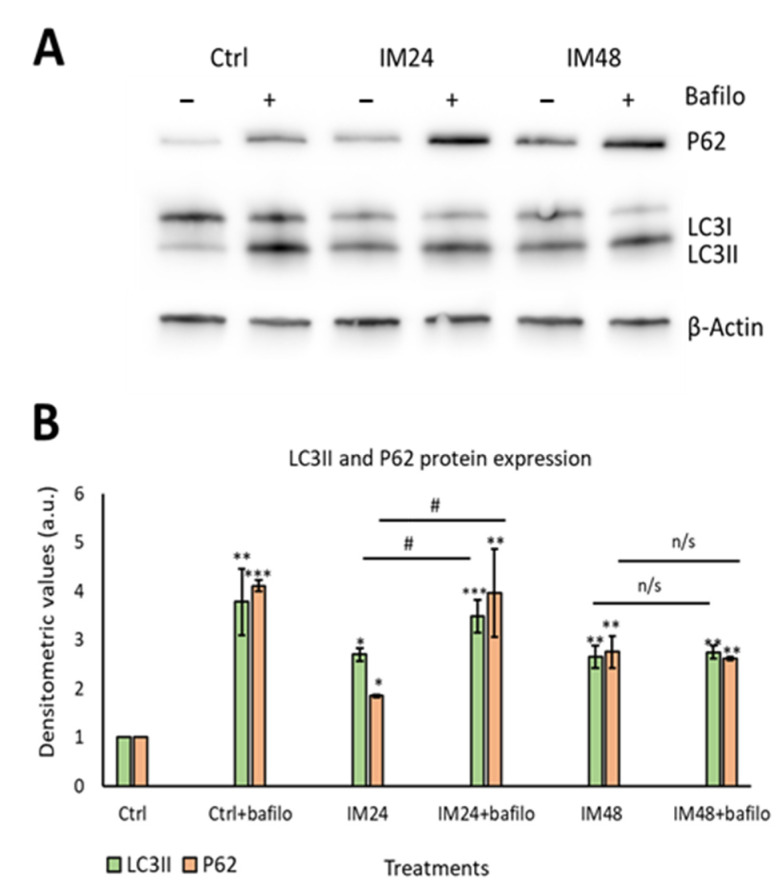
IM impairs the autophagic flux in hCPCs. (**A**) Representative image of Western blotting for P62 and LC3II after IM treatment with/without bafilomycin. (**B**) Quantification of Western blotting for LC3II (green) and P62 (orange). Data are mean ± SEM, *n* = 5; * *p* < 0.05, ** *p* < 0.05, and *** *p* < 0.001 vs. control; # *p* < 0.05 vs. IM 24 h treatment. n/s = not significant.

**Figure 5 ijms-23-11812-f005:**
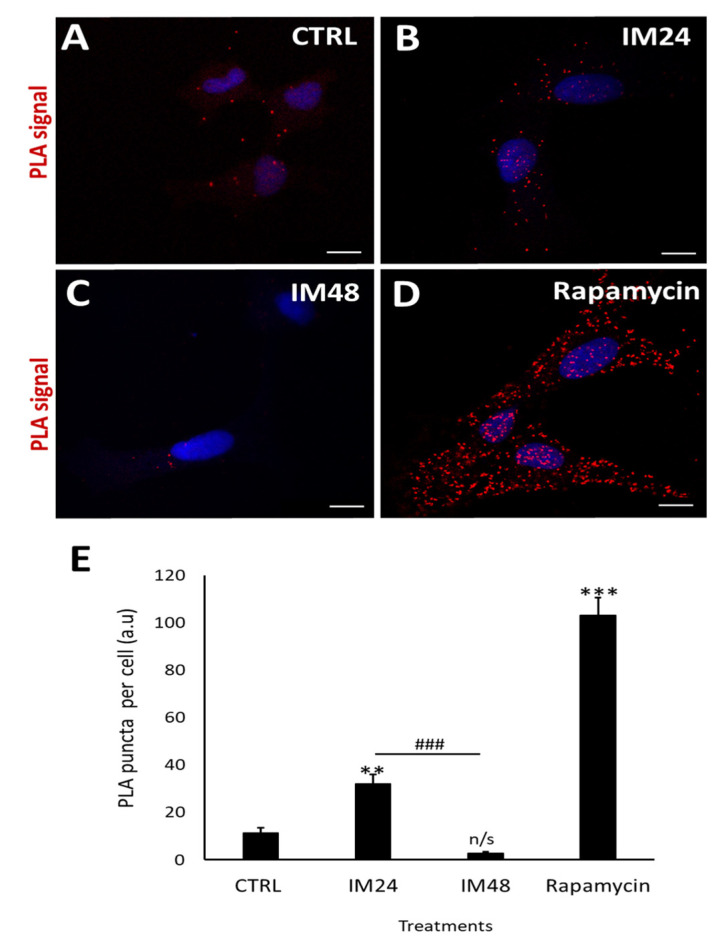
IM reduces LC3/p62 colocalisation in hCPCs. (**A**–**D**) PLA-representative images of control cells (low PLA signal), IM 24 h increased signal, IM 48 h reduced signal, and rapamycin high-PLA red puncta (arrows). (**E**) Quantification of PLA puncta after treatment with: IM 24 h, IM 48 h, and rapamycin 4 h. Data are mean ± SEM, *n* = 10; ** *p* < 0.05 and *** *p* < 0.001 vs. control; ### *p* < 0.001 vs. IM 24 h treatment. Scale bars = 10 µm. n/s = not significant vs. control.

**Figure 6 ijms-23-11812-f006:**
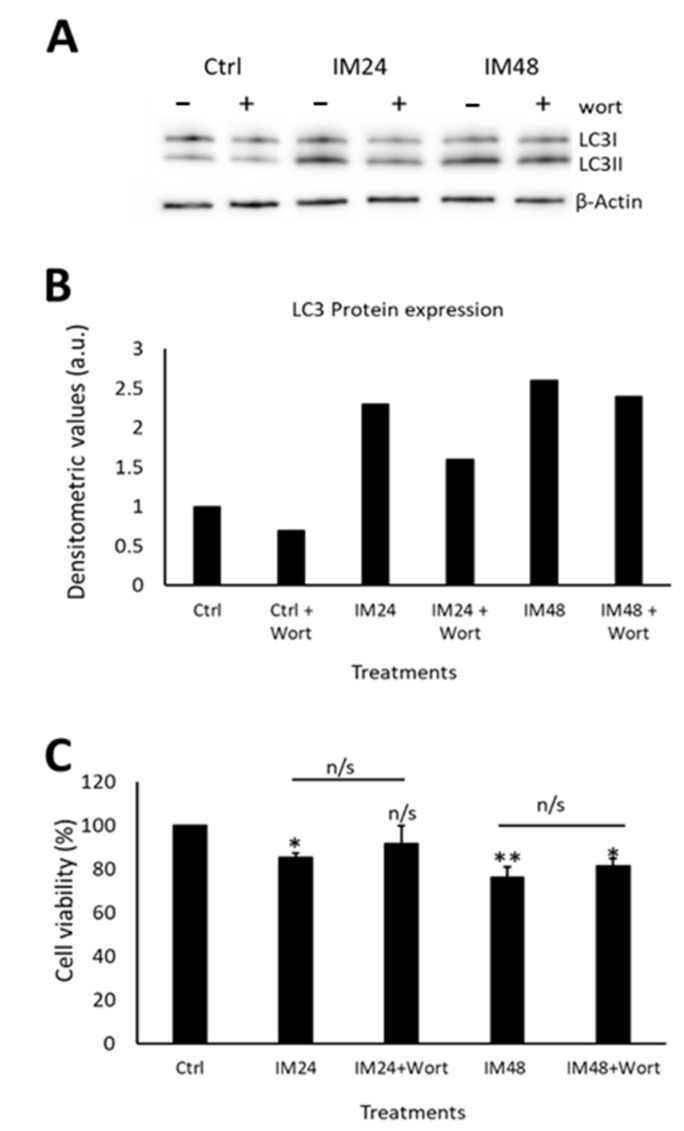
Autophagic flux only partially contributes to IM toxicity in hCPCs. (**A**) Western blot image and (**B**) densitometric values show LC3II expression after IM treatment for either 24 or 48 h, with/without wortmannin (wort) co-treatment, *n* = 1. (**C**) CPCs viability after treatment with or without IM and wortmannin treatment for 24 h. Data are mean ± SEM, *n* = 3; * *p* < 0.05 and ** *p* < 0.01 vs. control; n/s = not significant.

**Figure 7 ijms-23-11812-f007:**
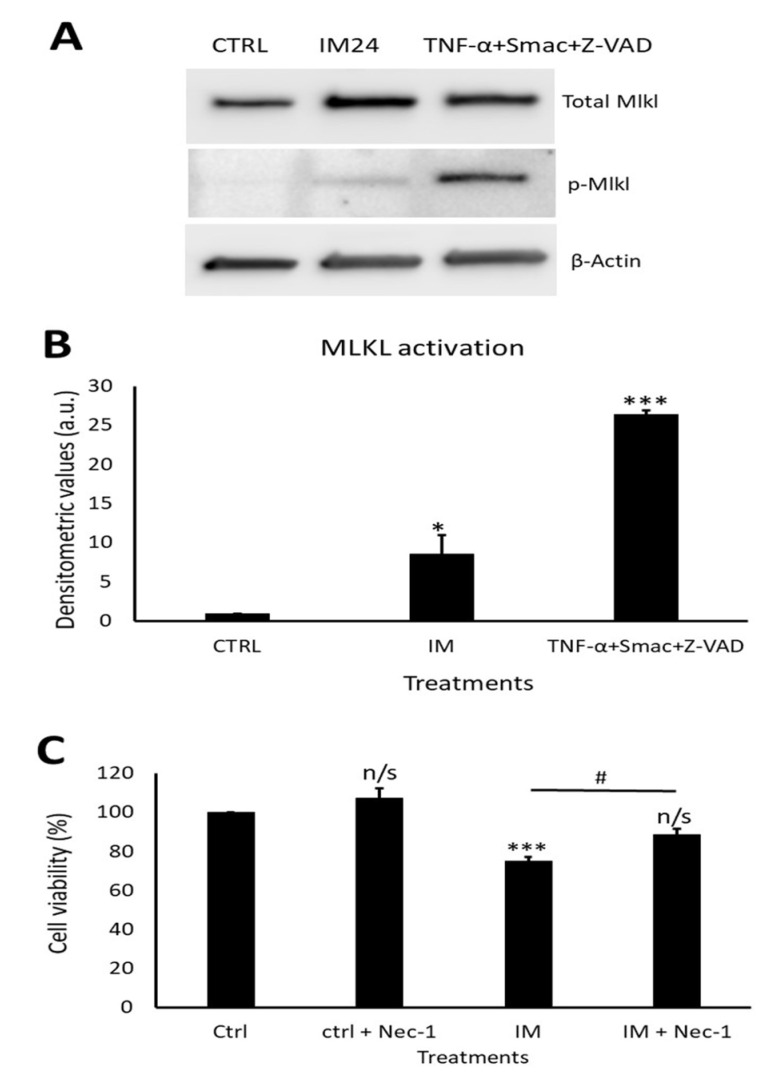
IM induces necroptosis cell death and is rescued by RIP1 inhibition. (**A**) Western blot representative image showing expression of total MLKL and p-MLKL after IM 24 h treatment and positive control. (**B**) Densitometric values show MLKL activation after IM treatment and in the positive control, *n* = 3. (**C**) Cell viability analysis of necrostatin (nec-1) inhibition with control cells treated with nec-1, IM alone, and IM co-treated with nec-1. Data are mean ± SEM, *n* = 4; * *p* < 0.05, *** *p* < 0.001, and n/s = not significant vs. control; # *p* < 0.05 vs. IM 24 h treatment.

**Figure 8 ijms-23-11812-f008:**
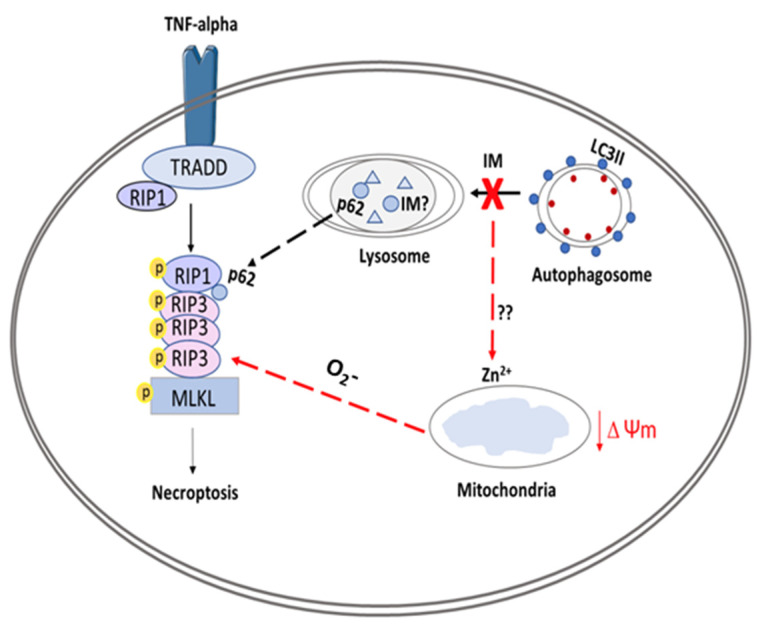
Schematic overview of proposed model for IM-induced cell death in CPCs. The figure shows IM being sequestered by the lysosomes, causing an impaired autophagic flux. This abnormal flux causes the release of zinc ions, which affect the Δψm, leading to production of superoxides (O_2_-); these superoxide ions can help to form the ripoptosome. The ripoptosome formation is also aided by leakage of p62 proteins from lysosomes and autophagosomes. Finally, the ripoptosome formation causes the activation/phosphorylation of MLKL and initiates necroptotic cell death.

**Table 1 ijms-23-11812-t001:** Table of primer pairs for real-time RT-qPCR. Forward and reverse primer sequences, product size, and mRNA accession number for each gene target.

Gene Targets	Forward Sequence (5′-3′)	Reverse Sequence (3′-5′)	Product Size (bp)	mRNA Accession Number
**β-Actin**	GCCTCGCCTTTGCCGA	CTCGTCGCCCACATAGGAAT	221	NM_001101.3
**BAX**	TAACATGGAGCTGCAGAGGATG	GGGACATCAGTCGCTTCAGTG	299	NM_001291428.1
**Calpain**	CAATGTCCGCTTCGGCTCTA	TGCGACTCACTGCGCC	182	NM_001749.3
**CASP8**	AGCCCTCTGAATTTGCTAGTC	AATATAATCCGCTCCACCCTTTCC	208	NM_001080125.1
**MLKL**	GTCCGGTACAGTCAGCAGAG	CTTCAAATTTTCCATGCCTTCGC	206	NM_152649.3
**PARP**	AGCGTGTTTCTAGGTCGTGG	CATCAAACATGGGCGACTGC	194	NM_001618.3
**RIP1**	CTGCTCGTCAAGTGTGGGA	CCGAGGTCTGCGATCTTAATGT	702	NM_003804.4
**RIP3**	TTACCTGCACGACCAGAACC	TGCTGCTCTTGAGCTGAGAC	548	NM_006871.3
**TNF-α**	TAGCCCATGTTGTAGCAAACCC	GGACCTGGGAGTAGATGAGGT	150	NM_000594.3

**Table 2 ijms-23-11812-t002:** Table of primary antibodies. List of antibodies used for Western blotting and ICC staining (including dilutions used and host species).

Host	Target	Company	Dilution
Rabbit	Anti-human LC3	Novus	1:4000
Rabbit	Anti-human MAP1LC3B	Cusabio	1:200
Mouse	Anti-human LAMP2	Novus	1:1000
Rabbit	Anti-human p-MLKL	Abcam	1:1000
Rabbit	Anti-human total MLKL	Genetex	1:1000
Rabbit	Anti-human p62	ProSci Inc.	1:1000
Rabbit	Anti-human beta-actin	Cell Signaling	1:2000

## Data Availability

Not applicable.

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
