# Peer review of "Imatinib Mesylate Induces Necroptotic Cell Death and Impairs Autophagic Flux in Human Cardiac Progenitor Cells"

_ijms, 2022, doi:10.3390/ijms231911812_

Round 1

Reviewer 1 Report

Summary

Imatinib is a receptor tyrosine kinase inhibitor that is well established anti-cancer agent, however it has been linked to cardiotoxicity. This study expanded on a previous study that looked at the effects of imatinib on adult rat cardiac fibroblasts to look at cardiac progenitor cells in humans (hCPCs). Using physiologically relevant concentrations of imatinib, Walmsley et al demonstrate that imatinib does indeed reduce hCPC viability in vitro. While there was no evidence of apoptosis after imatinib treatment, the authors did show an increase in phosphorylated MLKL, indicating necroptosis, and that imatinib-induced cell death could be rescued with Necrostatin-1. Imatinib also blocked autophagosome turnover, however this only had minimal effects on cell viability, as inhibition of autophagy with wortmannin did not rescue the cell death phenotype.

General Comments

Overall, this article was straight forward and well written. The autophagy and necroptosis results were very convincing. My only concern is regarding the apoptosis data. The images of the caspase 3/7 assay were difficult to interpret using the color scheme provided (red versus pink). Showing the data in individual panels would help. It would also be useful to see a positive control for apoptosis, like how a positive control was used for the PLA assay. I also think it is critical to have some sort of quantitative analysis of apoptosis (e.g., using annexin V flow cytometry, PARP cleavage western blot, luminescence caspase 3/7 microplate assay) to show more clearly that there is no apoptosis.

Specific Comments

Figure 1: Please clarify in the figure legend or text that Figure 1A was determined using FDA staining. Please refer to my general comments for additional experiments for this figure to show there is no apoptosis.

Author Response

General Comments

Overall, this article was straight forward and well written. The autophagy and necroptosis results were very convincing. My only concern is regarding the apoptosis data.

We thank the reviewer for their review and for their complimentary remarks about our manuscript.

The images of the caspase 3/7 assay were difficult to interpret using the color scheme provided (red versus pink). Showing the data in individual panels would help. It would also be useful to see a positive control for apoptosis, like how a positive control was used for the PLA assay.

We have re-presented these data, using individual panels as well as the combined whole image; with TMRM marking healthy cells in green, ToPro3 in pink, caspase-3/7 in red and DAPI in blue. The positive control data mentioned are now also present, as part of the data added to Figure 1 to address the following comment.

I also think it is critical to have some sort of quantitative analysis of apoptosis (e.g., using annexin V flow cytometry, PARP cleavage western blot, luminescence caspase 3/7 microplate assay) to show more clearly that there is no apoptosis.

We have carried out Annexin V-FITC labelling followed by flow cytometry analysis of the cells, using staurosporine as our positive control, with CPCs from the same preparation treated for the same duration. We confirmed that treatment with IM at doses of 10 µM or 100 µM did not cause apoptosis (consistent data were also found for a third sample treated with 50 µM IM). Annexin V staining was found in 2.27% of untreated cells, 2.05% of 10 µM IM-treated cells, 2.82% 100 µM IM-treated cells, compared to 95.53% of cells exposed to 10 µM staurosporine. These data have been added to the revised Figure 1.

Specific Comments

Figure 1: Please clarify in the figure legend or text that Figure 1A was determined using FDA staining.

The Results section text now includes this specific point.

Reviewer 2 Report

The authors report new and somewhat diverse data on putative unwanted effects of imatinib on human cardiac progenitor cells in vitro. I have following minor comments:

What was chemical purity of imatinib? Any knowledge about chemical impurities? Only highly pure compounds are studied in clinical trials. Palladium is often used as catalyst in various chemical reactions and it may be present as impurity in commercially available compounds. Further, palladium or other catalysts may have effects of their own to measured parameters such as mitochondrial function. Millimolar concentration of EDTA or EGTA can be used to chelate putative metal contaminants provided that additional calcium and magnesium is added to match physiologically relevant free concentrations.

Imatinib has pharmacologically active metabolite(s). Please clarify why only parent compound was studied here.

Mitochondrial uncoupling and proton leak by known reference compounds such as CCCP and FCCP were recently reported to be caused by activation of ADP/ATP carrier (Bertholet et al Nature PMID: 35614225). Does imatinib activate ADP/ATP carrier? Please discuss or preferably provide new data.

Mitochondrial uncoupling by small molecule dinitrophenol is known to induce hyperthermia in vivo. Is there any clinical data to support the idea that imatinib may induce mitochondrial uncoupling and hyperthermia in vivo?

Please omit P values from the abstract.

Please add journal information to reference #11.

Author Response

The authors report new and somewhat diverse data on putative unwanted effects of imatinib on human cardiac progenitor cells in vitro.

We thank the reviewer for their positive comments on our manuscript.

I have following minor comments:

What was chemical purity of imatinib? Any knowledge about chemical impurities? Only highly pure compounds are studied in clinical trials. Palladium is often used as catalyst in various chemical reactions and it may be present as impurity in commercially available compounds. Further, palladium or other catalysts may have effects of their own to measured parameters such as mitochondrial function. Millimolar concentration of EDTA or EGTA can be used to chelate putative metal contaminants provided that additional calcium and magnesium is added to match physiologically relevant free concentrations.

We have checked the chemical purity of the imatinib with the supplier and it has been confirmed to us as being determined (by HPLC) to be 99.8% pure. While the addition of EDTA to chelate the palladium would deal with this impurity risk, this could also lead to the chelation of zinc for example (Nyborg et al., 2004, The Biochem. J. 381(3):e3-4). Zinc removal can have detrimental impacts upon the phenotype of stem cells, affecting both pluripotent and adult stem cells (Mnatsakanyan et al. 2019, Front. Cell Dev. Biol. 7:180; Moon et al., 2018, Stem Cells Int. 5736535). Therefore, it would appear unlikely that the chelation of possible contaminants could be achieved without causing other, more significant, disruptions to cations’ effects on the very delicately balanced phenotype of stem cells. The issue of potential contaminants is nevertheless a potential study limitation and we have acknowledged this within the discussion.

Imatinib has pharmacologically active metabolite(s). Please clarify why only parent compound was studied here.

This study was part of an examination of three separate RTKIs, in which we characterised the cell death pathways of each: while we would have certainly been interested in examining the impacts of their pharmacologically active metabolites, there was neither funding nor time sufficient to allow us to cover every direction of subsequent investigation. However, the use of only the parent compound is another study limitation and we have now acknowledged this within the discussion.

Mitochondrial uncoupling and proton leak by known reference compounds such as CCCP and FCCP were recently reported to be caused by activation of ADP/ATP carrier (Bertholet et al. Nature PMID: 35614225). Does imatinib activate ADP/ATP carrier? Please discuss or preferably provide new data.

We cannot find any direct evidence of IM affecting this, although we have identified a paper that mentions imatinib and the uncoupling protein (UCP)-1. However there does not seem to be an effect, with no change in gene expression of UCP-1 seen in primary adipocytes following 10uM IM for 24 hours (Sylow et al., 2016, Endocrinology 157(11):4184-4191). We have added comments into the manuscript discussion regarding this point.

Mitochondrial uncoupling by small molecule dinitrophenol is known to induce hyperthermia in vivo. Is there any clinical data to support the idea that imatinib may induce mitochondrial uncoupling and hyperthermia in vivo?

We have carried out a thorough search of the literature and cannot identify any evidence linking imatinib with hyperthermia. While there were certainly papers containing both terms, in these cases the hyperthermia either referred to (pre-treatment) presenting symptoms or to exogenously applied hyperthermic anti-cancer treatment. On the contrary, one veterinary case report (Lee et al., 2022, Front. Vet. Sci. 8:625527) found that increasing imatinib treatment was associated with resolution of pre-existing CLL-associated hyperthermia.

Please omit P values from the abstract.

Please add journal information to reference #11.

These are both now amended, with thanks again to the reviewer for their evident diligence, in identifying our error of omission in the references.

Round 2

Reviewer 1 Report

I appreciate the changes made by the authors and believe it greatly improves the overall quality of the manuscript.

Author Response

We thank the reviewer again for their review and comments on our manuscript.